# PET-Uptake in Liver Metastases as Method to Predict Tumor Biological Behavior in Patients Transplanted for Colorectal Liver Metastases Developing Lung Recurrence

**DOI:** 10.3390/cancers14205042

**Published:** 2022-10-14

**Authors:** Svein Dueland, Tor Magnus Smedman, Harald Grut, Trygve Syversveen, Lars Hilmar Jørgensen, Pål-Dag Line

**Affiliations:** 1Experimental Transplantation and Malignancy Research Group, Division of Surgery, Inflammatory Diseases and Transplantation, Oslo University Hospital, 0424 Oslo, Norway; 2Section for Transplantation Surgery, Department of Transplantation Medicine, Oslo University Hospital, 0424 Oslo, Norway; 3Department of Oncology, Oslo University Hospital, 0424 Oslo, Norway; 4Institute of Clinical Medicine, University of Oslo, 0424 Oslo, Norway; 5Department of Radiology, Vestre Viken Hospital Trust, 3004 Drammen, Norway; 6Department of Radiology and Nuclear Medicine, Oslo University Hospital, 0424 Oslo, Norway; 7Department of Thoracic Surgery, Oslo University Hospital, 0424 Oslo, Norway

**Keywords:** liver transplantation, colorectal cancer, liver metastases, lung metastases, overall survival, fluorine 18 fluorodeoxyglucose (18F-FDG) positron emission tomography, pulmonary resection

## Abstract

**Simple Summary:**

Many colorectal cancer patients with liver-only metastases receiving liver transplantation develop pulmonary metastases after liver transplantation. Pre-transplant PET liver uptake determines overall survival in patients treated by post-transplant resection of pulmonary metastases.

**Abstract:**

The objective of the study was to determine the impact of PET uptake on liver metastases on overall survival (OS) after resection of pulmonary metastases in patients who had received liver transplantation (LT) due to unresectable colorectal liver-only metastases. Resection of pulmonary colorectal metastases is controversial. Some hospitals offer this treatment to selected patients, whereas other hospitals do not perform the procedure in colorectal cancer patients who develop pulmonary metastases. All patients included in the LT studies who developed pulmonary metastases as first site of relapse, and had resection of these as first treatment, were included in this report. Metabolic tumor volume (MTV) in liver was derived from the pre-transplant PET examinations. OS from time of resection was calculated by the Kaplan–Meier method. Patients with low MTV (<70 cm^3^) had significantly longer OS from time of resection of pulmonary metastases compared to patients with high MTV (>70 cm^3^). Patients with low MTV in the liver had 10-year OS from time of pulmonary resections of 86%. Liver MTV values from pre-transplant PET examinations may predict long OS in colorectal cancer patients with a resection of pulmonary metastases developing after LT. Thus, in selected colorectal cancer patients developing pulmonary metastases resection of these metastases should be the treatment of choice.

## 1. Introduction

Colorectal cancer (CRC) is a common malignancy worldwide and the second most frequent cause of cancer-related death in Western societies [1]. About half of the CRC patients have metastatic disease at time of diagnosis or in later stages of the disease. Liver is the most frequent metastatic site [2] followed by the lungs [3]. Liver resection (LR) due to colorectal liver metastases (CRLM) has been considered the most important curative treatment option with reported 5-year overall survival (OS) of 30–50% in most studies [4,5]. However, only a minority of patients with CRLM are candidates for LR [6]. Resection of pulmonary metastases in CRC patients is more controversial compared to LR. Many centers will resect pulmonary metastases in selected patients, after observation over time, whereas others argue that resection of pulmonary metastases will not improve survival [7,8,9,10]. 

Most CRC patients with liver-, lung- or other sites metastases receive palliative chemotherapy achieving a median OS of about 24–30 months from start of first line treatment [11,12,13]. However, longer OS may be obtained in selected patients with ECOG 0–1, RAS and BRAF wild type tumors and left-sided primary tumor location [14,15]. Median OS from start of second- and third-line chemotherapy is 10–12 and about 7 months, respectively [16,17,18].

Liver transplantation (LT) is the standard of care treatment for selected patients with acute and chronic liver failure as well as for selected patients with hepatocellular carcinoma (HCC), liver metastases from low-grade neuroendocrine tumors [19,20] and patients with hilar cholangiocarcinomas after neo-adjuvant chemo-radiation therapy [21]. In 2006 a pilot study (SECA-I) reexamining LT for unresectable CRLM was initiated at Oslo University Hospital. In the first report of this study Kaplan–Meier-calculated 5-year OS was 60% [22]. In a more recent study with more strict selection criteria (SECA-II study), a 5-year OS of 83% was reported [23]. In both studies the majority of the recurrences were lung metastases [24]. However, despite of short disease free survival, the implementation of strict selection criteria yields 5-year OS rates of up to 70–100% [25]. The importance of stringent patient selection is clearly illustrated by the low OS obtained in patients transplanted according to wide and liberal inclusion criteria [26]. In this context, patients with a primary tumor in the ascending colon have inferior survival prospects and should not be considered for LT [27].

The International Hepatico-Panreatico-Billiary Association (IHPBA) has recently published consensus guidelines for LT in CRC patients [28], and several transplant centers in the United States are now offering LT to highly selected patients with CRLM, including patients with liver failure after chemotherapy and without evidence of malignant disease. In addition, several ongoing clinical trials are investigating LT in non-resectable CRLM both in Europe and North America [29]. 

Unlike HCC patients, CRC patients developing relapse after LT may obtain long OS from the time of recurrent disease [25,30]. The present report describes the impact of pre-transplant PET-liver uptake on survival outcome in patients resected for pulmonary recurrence after LT for CRLM. 

## 2. Materials and Methods

Since initiating the SECA-I study in November 2006 re-examining LT in CRC patients, the concept of LT in CRC has been extended by several studies at Oslo University Hospital with different inclusion and exclusion criteria [22,23,26]. All patients included in the different prospective LT studies had signed an informed consent and all studies were approved by the Regional Ethics Committee and Institutional Review Board. The included patients were considered to have unresectable CRLM by the institutional multidisciplinary liver team. All patients had received 1–3 lines of chemotherapy prior to inclusion.

The different Clinicaltrail.gov registration numbers are: NCT01311453 (SECA-I study) and NCT01479608 (SECA-II). The inclusion and exclusion criteria for the different LT studies, as well as immunosuppression used in the different studies, have previously been reported [22,23,26]. Fluorine 18 fluorodeoxyglucose (18F-FDG) positron emission tomography in combination with computed tomography (PET/CT) was performed on all patients to exclude patients with extra-hepatic disease. Metabolic tumor volume (MTV) from the CRLM of each patient was obtained from the pre-operative PET scans as previously described [31]. None of the patients received adjuvant chemotherapy after LT. The patients had regular outpatient follow-up once a month the first year, every three months the second year and every six months thereafter. CT scans were performed every three months during the first two years. Treatment at time of relapse was at the discretion of the physician responsible for the treatment of the patients. Patients who developed one or a few technically resectable pulmonary metastases after LT were considered for resection when tumor diameter on CT scans reached 10–15 mm. At the time of pulmonary resection, the patients had a repeated PET/CT scan to rule out extra-pulmonary disease. No neoadjuvant/adjuvant chemotherapy was given before or after pulmonary resection. 

Disease-free survival (DFS) was defined as time from LT to suspected metastases or local relapse described by CT/MRI/PET-CT scans. OS was calculated from the date of LT until death or end of follow-up (1 August 2020). Survival from time of relapse was calculated as OS minus DFS in patients with recurrent disease. Survival time from pulmonary resection was calculated from the date of the lung resection until death or end of follow-up (1 August 2020).

Risk stratification was performed using the Fong Clinical Risk Score (FCRS) [32] and the Oslo Score (giving 1 point for each of the following pre-transplant characteristics: largest liver lesion >5.5 cm, plasma CEA levels >80 µg/L, time from surgery of primary tumor to LT of less than 24 months, progressive disease on chemotherapy at time of LT) [22] and PET MTV < 70 cm^3^.

### Statistical Analyses

Survival analyses were performed using the Kaplan–Meier method. The log-rank test was used to compare outcome between groups. Difference between median values of groups was calculated by the non-parametric Mann–Whitney U test. A *p*-value less than 0.05 was considered statistically significant. The analyses were performed by IBM SPSS Statistics for Windows, version 25.0, IBM Corp, Armonk, NY, USA.

## 3. Results

A total of 55 CRLM patients were included in prospective LT studies at Oslo University Hospital between November 2006 and 1 August 2020. At end of follow-up, 25 patients had pulmonary metastases as the first site of relapse and 14 of these underwent pulmonary resection as the first treatment of relapse after LT. These 14 patients received LT from November 2006 to November 2017, and resection of pulmonary metastases from February 2008 to January 2020. All pulmonary resected samples were confirmed histologically as CRC metastases. In addition, one patient also received resection of a primary non-small cell carcinoma. Baseline characteristics at the time of LT for the 14 patients are given in Table 1. The eleven other patients with pulmonary metastases as first site of relapse received palliative chemotherapy (*n* = 5), palliative radiation therapy (*n* = 1), radiofrequency ablation therapy (*n* = 1), three patients received resection of other metastatic sites as first treatment and one patient has not started any treatment yet. The five patients that were only offered palliative chemotherapy had five to more than 10 pulmonary metastases at time of relapse, with sizes ranging from 3 to 22 mm.

Median DFS from time of LT in the 14 patients treated by curative intended lung resection was 12.0 months (95% CI 11.2–12.7 months, Figure 1A) with 12 of 14 with DFS for less than 24 months. At time of diagnosis of pulmonary metastases by CT scans, two patients had two pulmonary metastases and 12 patients had one metastasis. One of the two patients with two metastases underwent resection of both metastases during the same procedure and the other patient underwent resections in two consecutive procedures. Median time from relapse to the first resection of pulmonary metastases was 8.1 months (95% CI 4.1–12.1 months) and median size at CT scans at time of diagnosis of recurrence was 6 mm (range 5–13 mm). A total of 22 pulmonary lesions were resected, and the median tumor volume doubling time was 132 days (range 35–282 days), median size at last CT scan before resection was 13 mm (range 7–32 mm) and finally, the median size at histological examination was 14.5 mm (range 5–31 mm).

Median OS after relapse has not been reached with 10-year survival of 68.8%. Median OS from the time of the first pulmonary resection in the 14 patients has not been reached with 10-year survival of 64.5% with median follow-up of patients alive after pulmonary resection of 98.3 months (Figure 1B). At end of the follow-up, six patients have been observed for more than five years after pulmonary resection. Eight patients have no evidence of disease (NED) 7–131 months after pulmonary resection. Of these eight patients, seven has been observed from 75–156 months, and the last patient for 32 months after LT. Six patients have received chemotherapy after resection of pulmonary metastases. Median time from resection of pulmonary lesions to start of palliative chemotherapy was 36.4 months (range 7.4–112.9 months). Five of these six patients are deceased, and one patient is alive with metastatic disease. Median OS from start of palliative chemotherapy in these six patients was 12.1 months (95% CI 9.5–14.7 months).

### 3.1. PET-Metabolic Tumor Volume in Liver 

It has previously been shown that patients transplanted for CRLM and with a MTV-liver uptake value <70 cm^3^ on the pre-transplant PET-scan have significant better OS and OS after relapse compared to patients with MTV values > 70 cm^3^ [31]. In the current sub-cohort of 14 patients treated with pulmonary resection as first treatment with curative intent, nine patients had MTV <70 cm^3^ (low) and five had MTV >70 cm^3^ (high). Patients receiving pulmonary resection as first treatment of relapse with low MTV-values (*n* = 9) had median DFS of 16.1 months (95% CI 3.9–28.3 months) vs. 9.7 months (95% CI 3.8–15.5 months) in patients with high MTV values (*n* = 5), respectively (*p* = 0.030 Figure 2A).

Patients with low MTV had 10-year OS from time of relapse of 88.9% compared to 26.7% in patients with high MTV liver values (*p* = 0.021, Figure 2B). OS from the time of pulmonary resection in these patients were 85.7% and 25.0% at 10 years in patients with low and high PET MTV values, respectively (*p* = 0.028, Figure 2C). There was no significant difference in size of the pulmonary metastases at time of diagnosis (*p* = 1.000), at time of resection (*p* = 0.116) or in growth rate (tumor volume doubling time) (*p* = 0.311). Finally, histological examination of the tumors revealed no size difference between the two groups (*p* = 0.303) (Table 2).

### 3.2. Fong Clinical Risk Score (FCRS, Oslo Score and GAME Score)

All patients with FCRS of 0–2 (*n* = 5) at time of LT are alive 42–145 months after resection of pulmonary metastases compared to OS at 5 and 10 years of 46.9 % in patients with FCRS of 3–5 (*p* = 0.108, Figure 3A). Patients with Oslo Score of 0–2 (*n* = 12) and Oslo Score of 3–4 (*n* = 2) at time of LT had 10-year OS after resection of pulmonary metastases of 76.2% and 0%, respectively (*p* < 0.001 Figure 3B). The GAME score of the 14 patients receiving resection of pulmonary metastases as first treatment was also calculated. Since patients with extra-hepatic disease were excluded from liver transplantation the maximum score would be 5. On these 14 patients median score was 3 (two patients each with a score of 1 and 2, seven patients with a score of 3 and three patients with a score of 4. The 5-year OS from time of relapse in these patients were 100% in patients with score of 1 or 2, 68.6% in patients with a score of 3 and 33.3% in patients with a score of 4 (*p* = 0.244).

### 3.3. Other Factors

Median OS from the time of pulmonary resections in patients with progressive disease on chemotherapy at time of LT was 58.2 months compared to not reached in patients with stable disease or response to chemotherapy (*p* = 0.114). In addition, there was no significant difference in OS after resection of pulmonary metastases related to KRAS status (*p* = 0.675) or pN0 vs. pN+ (*p* = 0.513) of the primary tumor.

## 4. Discussion

Liver and lungs are the most frequent sites of metastatic disease in CRC patients and the lungs are a frequent site of metastatic disease following LR [33]. LR is considered standard of care in resectable CRLM. Given a limited survival probability for patients on palliative chemotherapy, borderline tumor resectability is commonly accepted at many hospital MDT-meetings [34], although no randomized study comparing LR to chemotherapy/radiation therapy has to our knowledge been reported. Several prognostic scoring systems for resection of CRLM have been developed, including Fong Clinical Risk Score [32]. In contrast to LR, resection of pulmonary CRC metastases is more controversial, with several centers offering this treatment option to selected patients whereas other hospitals do not offer this treatment. A randomized study with pulmonary resection in selected CRC patients has been initiated but was closed due to slow accrual [10].

We have previously shown that LT results in higher OS in CRLM patients with high tumor load compared to liver resected patients [27,35]. CRLM patients with favorable prognostic factors seem to have a comparable 5-year OS rate to conventional indications for LT, including patients with HCC within the established Milan or Metro-ticket 2.0 criteria [30,36]. Due to the scarcity of donor grafts LT can only be a treatment option in a minority of patients with CRLM. Several studies with LT in CRC patients are ongoing in Europa and Canada, and several US-transplant centers have now transplant programs considering selected CRC patients for LT [29]. It is therefore important that the international medical community is aware of LT as a possible treatment option for highly selected CRLM patients.

In accordance with other publications, we have previously reported a Kaplan–Meier calculated 5-year OS of 69% in rectal cancer patients treated with resection of pulmonary metastases with median OS of 71 months [37]. In comparison, patients receiving palliative care for lung metastases had 5-year OS of 11% with median OS of 22 months and no patient survived beyond 80 months [37]. However, as observed in the present report, patients receiving resection of the pulmonary metastases had a median of one metastasis at time of relapse after LT compared to more than 10 lesions in patients starting palliative treatment.

We have previously shown that CRC patients with a low (<70 cm^3^) PET-MTV liver value at pre-transplant examination have improved OS from time of relapse after LT compared to patients with high PET-MTV [25]. In the present report we show that pre-transplant PET-MTV liver values also predict survival after lung resection post-LT, often performed more than a year after transplant. This may seem surprising given that there were no differences in size of the metastatic lesions at any time point (Table 2), which makes it relevant to hypothesize that liver PET-MTV represents a robust surrogate marker of the “tumor biology” that may be useful in clinical decision making if recurrence of the malignant disease is detected. PET-MTV in CRLM may also be of importance in the treatment of relapse after LR and not only after LT [38]. Consequently, PET-CT examination determining MTV in CRLM might be incorporated in the work-up also for patients scheduled for LR.

Despite the lack of randomized studies on pulmonary resection in CRC patients, the present results with an estimated 10-year OS rate of more than 85% suggest that surgery with curative intent should be offered to CRC patients with few and small pulmonary lesions that have been observed over time without other metastatic sites. Albeit the present study is a very small one, we are not aware of any report of similar or close to a 10-year OS rate after chemotherapy for pulmonary metastases in CRC patients. Stereotactic body radiation therapy (SBRT) has been reported to have relative similar OS compared to resection in patients with small and few lesions in non- small cell lung cancer [39], so SBRT might also be a treatment option in CRC patients with pulmonary metastases, especially in case of borderline resectable or inoperable patients.

## 5. Conclusions

In conclusion, PET- MTV liver uptake predicts OS after resection of pulmonary metastases developing after LT for CRLM and patients with low PET-MTV should be offered surgery with a curative intent whenever possible. It is reasonable to assume that PET-MTV liver uptake also will predict OS after resection of CRC pulmonary metastases in patients receiving LR. PET scans might be considered as part of the standard work-up before surgical treatment of CRLM.

## Figures and Tables

**Figure 1 cancers-14-05042-f001:**
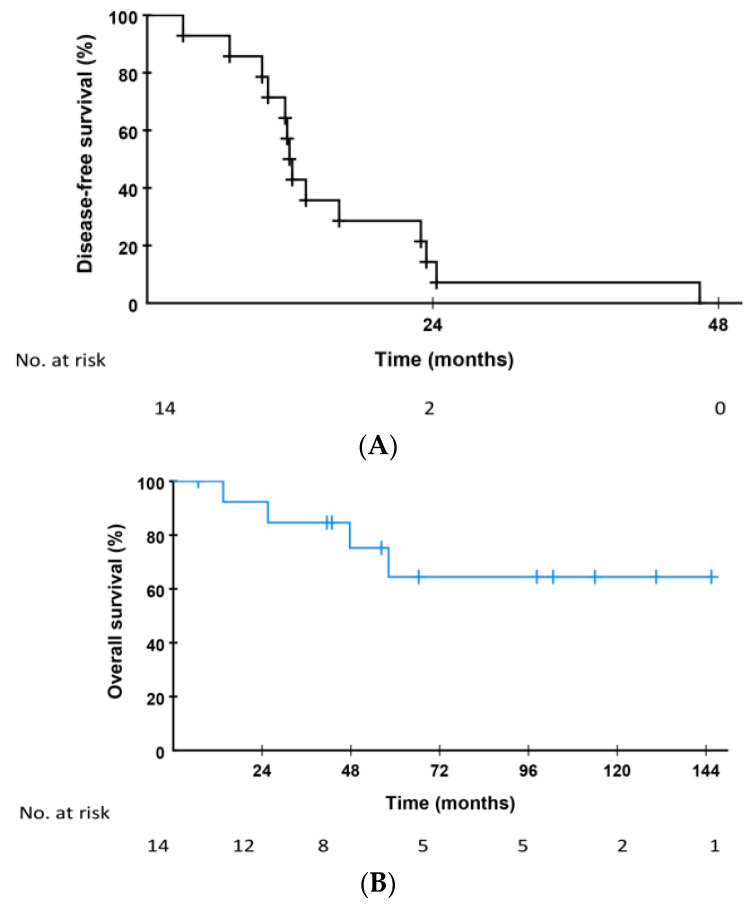
(**A**) Disease-free survival from time of liver transplantation (LT) in the 14 patients who had lung metastases as first site of relapse after LT and received resection of the pulmonary lesion (s) as first treatment. (**B**) Overall survival from time of pulmonary resections in the 14 patients who had lung metastases as first site of relapse after LT and received resection of the pulmonary lesion (s) as first treatment.

**Figure 2 cancers-14-05042-f002:**
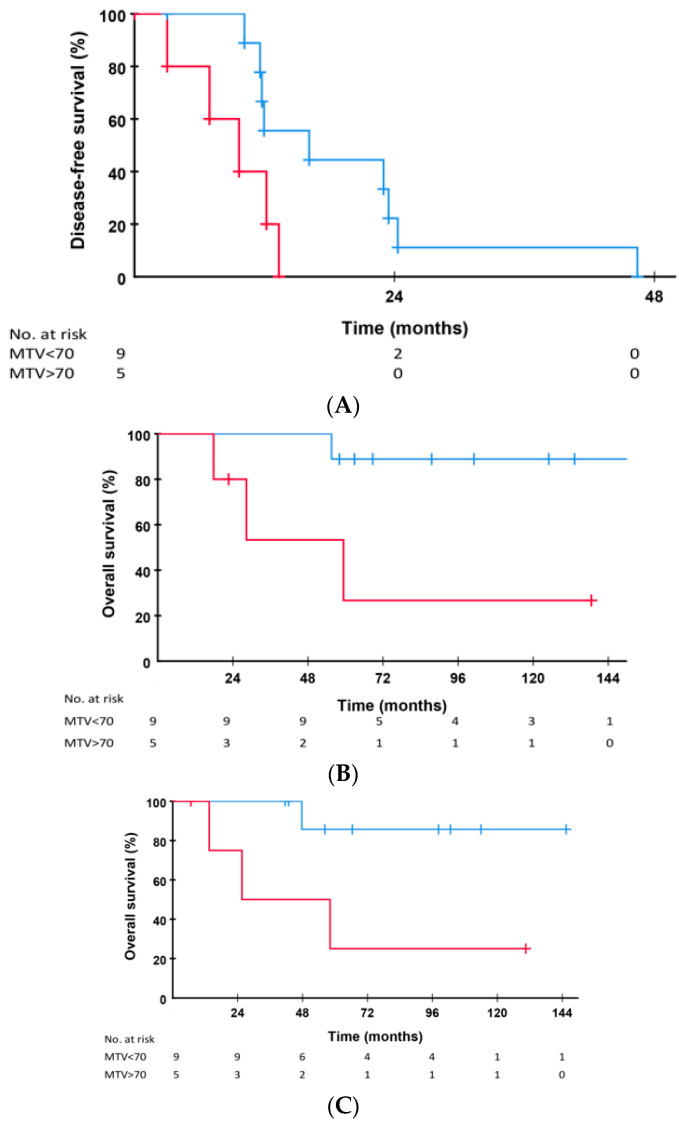
(**A**) Disease-free survival from time of liver transplantation (LT) in the 14 patients who had lung metastases as first site of relapse after LT and received resection of the pulmonary lesion (s) as first treatment. Patients ith low liver PET metabolic volume (MTV) (*n* = 9, blue line) versus patients with high MTV values (*n* = 5, red line, *p* = 0.063). (**B**) Overall survival from time of relapse in the 14 patients who had lung metastases as first site of relapse after LT and received resection of the pulmonary lesion (s) as first treatment. Patients with low liver PET metabolic volume (MTV) (*n* = 9, blue line) versus patients with high MTV values (*n* = 5, red line, *p* = 0.013). (**C**) Overall survival from time of pulmonary resection in patients with low (*n* = 9, blue line) versus high MTV-values (*n* = 5, red line, *p* = 0.023).

**Figure 3 cancers-14-05042-f003:**
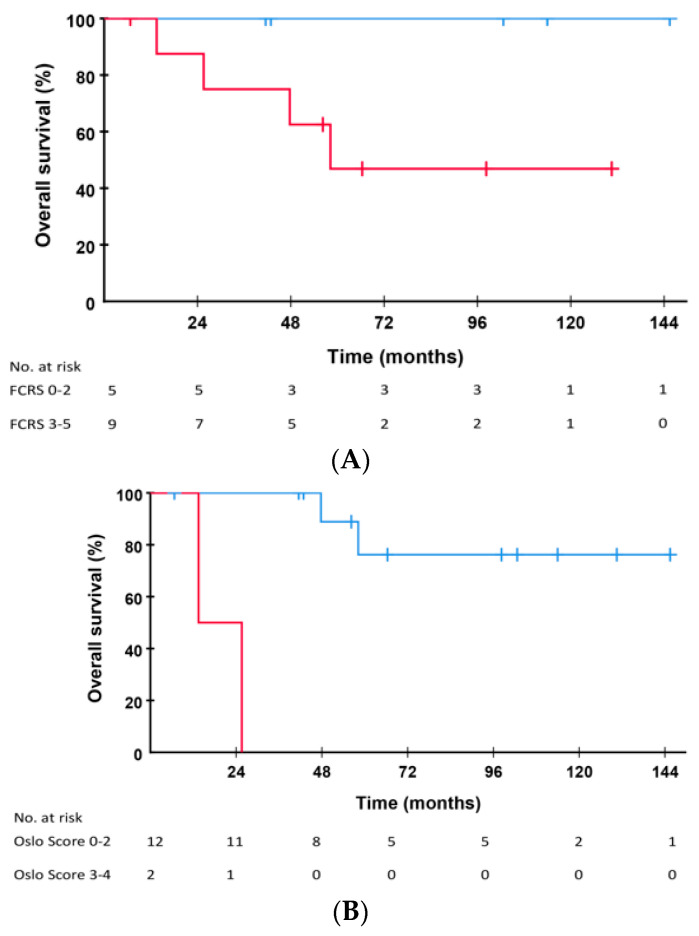
(**A**) Overall survival from time of pulmonary resection in patients with Fong Clinical Risk Score of 0–2 (*n* = 5, blue line) versus FCRS of 3–5 (*n* = 9, red line, *p* = 0.108). (**B**) Overall survival from time of pulmonary resection in patients with Oslo Score of 0–2 (*n* = 12, blue line) versus Oslo Score of 3–4 (*n* = 2, red line, *p* < 0.001).

**Table 1 cancers-14-05042-t001:** Baseline characteristics of the 14 patients receiving resection of pulmonary metastases as first treatment after liver transplantation and the 11 patients receiving other treatments.

	Lung Resection *n* = 14	Other Treatments *n* = 11	*p*-Values *
Age, median (range)	54. years (28.7–64.2 years)	58.2 years (42.0.70.0 years)	0.406
Sex	11 male/3 female	5 male/6 female	0.087
ypT (0/1/2/3)	1/0/3/10	0/1/2/6/2	0.297
ypN (0/1/2)	7/3/4	2/3/6	0.239
Location of primary tumor	Ascending colon 2, left colon 3, sigmoid 2, rectal 7	Ascending colon 5, left colon 1, sigmoid 4, rectal 1	0.066
Prior lines of chemotherapy	1. line 5, 2. line 7, 3. line 2	1. line 1, 2. line 9, 3. line 1	
Progressive disease at LT	No = 9, Yes = 5	No = 6, Yes = 5	0.622
KRAS mutant	No = 9, Yes = 5	No = 6, Yes = 5	0.622
Number of liver metastases (median, range)	7 lesions (5–40 lesions)	20 lesions (1–40 lesions)	0.126
Size of largest liver metastases (median, range)	37 mm (10–105 mm)	45 mm (3–130 mm)	0.247
Plasma CEA values µg/L (median, range)	5.0 (1–671)	10.0 (1–4346)	0.462
PET-liver MTV value (median, range)	32.5 cm^3^ (0–387 cm^3^)	61.6 cm^3^ (0–397 cm^3^)	0.453
Fong Clinical Risk Score (median, range)	3 (1–5)	4 (2–5)	0.288
Oslo Score (median, range)	1 (0–4)	2 (0–4)	0.653

Abbreviations: LT, liver transplantation; KRAS, Kirsten rat sarcoma virus; CEA, carcinoembryonic antigen; PET, positron emission topography; MTV, metabolic tumor volume. * Chi square.

**Table 2 cancers-14-05042-t002:** Largest diameter on CT scans at time of diagnosis and time of resection of pulmonary metastases, doubling- time of pulmonary metastases and largest diameter at histology.

	Liver PET MTV < 70 cm^3^	Liver PET MTV > 70 cm^3^	*p*-Value
Diameter at diagnosis (median, range)	7 mm, 5–10 mm	6 mm, 9–15 mm	*p* = 0.301
Diameter at resection (median, range)	13 mm, 7–14 mm	13 mm, 9–15 mm	*p* = 0.580
Doubling-time (median, range)	196 days, 77–282 days	104 days, 35–172 days	*p* = 0.266
Size at histology (median, range)	10 mm, 8–17 mm	14 mm, 9–23 mm	*p* = 1.000

Abbreviations: PET, positron emission topography; MTV, metabolic tumor volume.

## Data Availability

The data are not available due legal and privacy restrictions.

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
