# Peer review of "PET-Uptake in Liver Metastases as Method to Predict Tumor Biological Behavior in Patients Transplanted for Colorectal Liver Metastases Developing Lung Recurrence"

_cancers, 2022, doi:10.3390/cancers14205042_

Round 1

Reviewer 1 Report

Dear authors,

I appreciate the opportunity to review this manuscript. The authors are dealing with a significant problem: tumour biological behaviour in patients transplanted for colorectal liver metastases. The manuscript is well written, and the findings and the clinical application of this study are welcome for clinicians. 

Despite this, I have two comments regarding the manuscript:

Comment 1. It would be interesting to see the differences in baseline characteristics between the 25 patients with pulmonary metastases as the first site or relapse and the 14 patients that underwent pulmonary resection and LT.

Comment 2. Please revise the Conclusions section. The manuscript appears to be empty (line 294), but the conclusions appear on line 288.    

Author Response

Responses Reviewer 1:

  1. Results from the 11 patients not receiving lung resection as first treatment after relapse as pulmonary metastases has been added to Table 1.
  2. The Conclusion heading has been adjusted.

Reviewer 2 Report

Thank you for allowing me to review this well-presented manuscript by Dueland and Line et al. Herein, the authors tried to evaluate the prognostic importance of PET-uptake of original liver metastases, which can predict prognosis after pulmonary resection for lung metastases. Overall, the manuscript is well written and well presented. I have some comments which may increase the strength of the study.

1.         If possible, can you please include patients who did not receive pulmonary resection? I believe those patients should have higher MTV

2.         Fong score was a gold standard in the past. However, the score was made before the introduction of effective chemotherapies. Please use other clinical risk scores such as GAME or modified CRS, including genetic information.

3.         There was no comparison between MTV and tumor morphology. It would be interesting if you could add some comparisons. 

Author Response

Responses Reviewer 2:

  1. The MTV values of patients receiving resection of pulmonary metastases as first treatment of relapse (n=14) and patients receiving other treatments (n=11) has been added to Table 1.
  2. We have calculated the GAME score of the 14 patients receiving resection of pulmonary metastases as first treatment. Since patients with extra-hepatic disease were excluded from liver transplantation the max score would be 5. On these 14 patients median score was 3 (two patients each having score of 1 and 2, seven patients having score of 3 and three patients having score of 4. The 5-year OS from time of relapse in these patients were 100% in patients with score of 1 or 2, 68,6% in patients having score of 3 and 33.3% in patients having score of 4 (p=0.244)
  3. The histological report from liver explant as well as pulmonary resection specimens did only confirm adenocarcinoma metastases from CRC without any further grading of differentiation. MTV-values could therefore not be related to morphology of the histology specimens.